# Quantum spin transistor with a Heisenberg spin chain

O.V. Marchukov[1,2], A.G. Volosniev[1,3], M. Valiente[4], D. Petrosyan[5] & N.T. Zinner[1]

Spin chains are paradigmatic systems for the studies of quantum phases and phase transitions, and for quantum information applications, including quantum computation and short-distance quantum communication. Here we propose and analyse a scheme for conditional state transfer in a Heisenberg *XXZ* spin chain which realizes a quantum spin transistor. In our scheme, the absence or presence of a control spin excitation in the central gate part of the spin chain results in either perfect transfer of an arbitrary state of a target spin between the weakly coupled input and output ports, or its complete blockade at the input port. We also discuss a possible proof-of-concept realization of the corresponding spin chain with a one-dimensional ensemble of cold atoms with strong contact interactions. Our scheme is generally applicable to various implementations of tunable spin chains, and it paves the way for the realization of integrated quantum logic elements.

[1] Department of Physics and Astronomy, Aarhus University, DK-8000 Aarhus C, Denmark. [2] Center for Theoretical Physics, Department of Physics and Astronomy, Seoul National University, Seoul 08826, Korea. [3] Institut für Kernphysik, Technische Universität Darmstadt, 64289 Darmstadt, Germany. [4] SUPA, Institute of Photonics and Quantum Sciences, Heriot-Watt University, Edinburgh EH14 4AS, UK. [5] Institute of Electronic Structure and Laser, FORTH, GR-71110 Heraklion, Crete, Greece. Correspondence and requests for materials should be addressed to D.P. (email: dap@iesl.forth.gr) or to N.T.Z. (email: zinner@phys.au.dk).

Starting with the original proposal by Datta and Das[1] for a spin-based field-effect transistor, the field of spintronics[2] has explored how the spin degrees of freedom can be used for information transfer. More than two decades later, this research has reached the quantum regime[3]. One motivation for this is the desire for miniaturization which led to the realization of single-electron transistors[4], or more generally single-dopant devices[5]. A second motivation is the potential applications in quantum information and computation. Two guiding proposals in the field involve implementation of quantum gate operations in quantum dots[6] and in doped silicon[7]. Shortly thereafter, molecular magnets have also been proposed[8]. It was subsequently shown that universal quantum computation can be realized with Heisenberg exchange interaction, known from quantum magnetism, alone[9,10].

Here we put forward a scheme for a quantum spin transistor that may serve as an integral component of quantum information devices. Similarly to the quantum computation proposals, it can be implemented with architectures that realize a Heisenberg spin chain. Various physical realizations of spin chains are being actively explored for short-range quantum state transfer required to integrate and scale-up quantum registers involving many qubits[11–15]. In fact, spin chains of the Heisenberg type have been realized in organic and molecular magnets[16], quantum dots[17], various compounds[16,18,19], Josephson junction arrays[20], trapped ions[21,22], in atomic chains on surfaces[23–27] and in thin films or narrow magnetic strips that carry spin waves[28,29]. Combined with conditional dynamics implementing quantum logic gates, spin chains can greatly facilitate large-scale quantum information processing. While many different implementations of the coherent spin transistor may be possible, here we focus on one such proof-of-concept realization with a small ensemble of strongly interacting cold atoms trapped in a tight one-dimensional potential of appropriate shape. Cold atoms have already been used to realize spin chains, and observations of Heisenberg exchange dynamics[30], spin impurity dynamics[31] and magnon bound states[32] have been reported.

## Results

**Coherent spin transistor.** Our quantum spin transistor works with an arbitrary spin state $|\psi\rangle = \alpha|\downarrow\rangle + \beta|\uparrow\rangle$ at the input port (target spin) which is coherently transferred to the output port, if there are no excited spins in the gate, $|0\rangle_{\text{gate}}$. However, if the gate contains an excited stationary spin (control spin), $|1\rangle_{\text{gate}}$, it completely blocks the transfer of the target spin state between the input and output ports. In other words, we have coherent dynamics for the initial state of the system $|\psi\rangle_{\text{in}}|0\rangle_{\text{gate}}|\downarrow\rangle_{\text{out}} \rightarrow |\downarrow\rangle_{\text{in}}|0\rangle_{\text{gate}}|\psi\rangle_{\text{out}}$ when the gate contains no spin excitation, but complete absence of dynamics for the state $|\psi\rangle_{\text{in}}|1\rangle_{\text{gate}}|\downarrow\rangle_{\text{out}}$ when the gate contains a single spin excitation. Our scheme thus realizes a quantum logic operation and it can be used to obtain spatially entangled states of target and control spins, as well as to create Schrödinger cat states for a large number of target spins.

The functionality of the proposed transistor relies on the ability to tune the energy levels of the gate such that one of its states is resonant with the input and output states, resulting in coherent excitation transfer, while the other state of the gate is non-resonant and is used to block the transfer. A system that offers sufficient flexibility to realize this level of control is the Heisenberg $XXZ$ spin-$\frac{1}{2}$ chain in a longitudinal magnetic field. We thus consider a chain of $N$ spin-$\frac{1}{2}$ particles described by the $XXZ$ model Hamiltonian ($\hbar = 1$)

$$H = \sum_{j=1}^{N} h_j \hat{\sigma}_z^j - \frac{1}{2}\sum_{j=1}^{N-1} J_j[\hat{\sigma}_x^j \hat{\sigma}_x^{j+1} + \hat{\sigma}_y^j \hat{\sigma}_y^{j+1} + \Delta \hat{\sigma}_z^j \hat{\sigma}_z^{j+1}], \quad (1)$$

where $\hat{\sigma}_{x,y,z}^j$ are the Pauli matrices acting on the $j$th spin, $h_j$ determine the energy shifts of the spin-up and spin-down states playing the role of the local magnetic field, $J_j$ are the nearest-neighbour spin–spin interactions, and $\Delta$ is the asymmetry parameter: $\Delta = 0$ corresponds to the purely spin-exchange $XX$ model, $\Delta = 1$ to the homogeneous spin–spin interaction $XXX$ model, while the limit of $|\Delta| \gg 1$ leads to the Ising model. We assume a spatially symmetric spin chain with $J_j = J_{N-j}$ and $h_j = h_{N+1-j}$ with $h_{1,N} = 0$. Note that we do not specify the sign of $J_j$, which, therefore, can be positive or negative.

The input and output ports for the target spin are represented by the first $j = 1$ and last $j = N$ sites of the chain, see Fig. 1a. The inner sites $j = 2, \ldots, N-1$ constitute the gate which may be open (Fig. 1b) or closed (Fig. 1c) for the target spin transfer depending on the absence, $|0\rangle_{\text{gate}}$, or presence, $|1\rangle_{\text{gate}}$, of a single control spin excitation. One might think that the shortest possible spin chain to accommodate a gate between the input and output ports would consist of just $N = 3$ spins. As we show in Supplementary Note 1, however, the three-spin chain cannot implement a reusable spin transistor even in the Ising limit $|\Delta| \gg 1$ since the control spin excitation at the gate site $j = 2$ is not protected from leakage. We will therefore illustrate the scheme using a chain of $N = 4$ spins, with the gate consisting of spins $j = 2,3$ coupled to each other via the strong exchange constant $|J_2| \gg |J_{1,3}|$.

Consider a system in the initial state $|\uparrow\downarrow\downarrow\downarrow\rangle \equiv |\uparrow\rangle_{\text{in}}|0\rangle_{\text{gate}}|\downarrow\rangle_{\text{out}}$, which we aim to efficiently transfer to the final state $|\downarrow\downarrow\downarrow\uparrow\rangle \equiv |\downarrow\rangle_{\text{in}}|0\rangle_{\text{gate}}|\uparrow\rangle_{\text{out}}$ using an intermediate resonant state, see Fig. 1b and Supplementary Note 2. The initial and final states have the same energy of $\tilde{\lambda}_{\uparrow\downarrow\downarrow\downarrow(\downarrow\downarrow\downarrow\uparrow)} = -\frac{1}{2}J_2\Delta - 2h$, where $h \equiv h_{2,3}$ (as is the standard convention in quantum optics, we refer to the expectation value of the Hamiltonian in a given state as the energy of this state). In turn, the eigenstates of Hamiltonian (1) in the single excitation space of strongly coupled sites $j = 2,3$ are given by $|G_{\pm}\rangle = \frac{1}{\sqrt{2}}(|\uparrow\downarrow\rangle \pm |\downarrow\uparrow\rangle)$, with the corresponding energies $\lambda_{\pm} = \frac{1}{2}J_2\Delta \mp J_2$ split by $2J_2$. Then, by a proper choice of the magnetic field, $h = h_{\pm} \equiv \pm\frac{1}{2}J_2(1 \mp \Delta)$, we can tune the energy of one of the intermediate states $|\downarrow G_{\pm} \downarrow\rangle$ into resonance with the initial $|\uparrow\downarrow\downarrow\downarrow\rangle$ and final $|\downarrow\downarrow\downarrow\uparrow\rangle$ states (for example, for $h = h_-$ the resonant state is $|\downarrow G_- \downarrow\rangle$). Simultaneously, the other intermediate state does not participate in the transfer since its energy is detuned by $2J_2$ which is much larger than the exchange coupling $J_1/\sqrt{2}$ of the initial and final states to the intermediate states. The transfer time of the spin excitation between the initial and final states via a single resonant intermediate state is $t_{\text{out}} = \pi/|J_1|$ (Fig. 1d).

Next, we place a single spin excitation in one of the eigenstates $|G_{\pm}\rangle$ of the gate. To be specific, for the magnetic field $h = h_-$ we place the control spin in state $|G_+\rangle$ and denote it as $|1\rangle_{\text{gate}} \equiv |G_+\rangle$. Then the control spin cannot leak out of the gate region and therefore it is stationary. Moreover, if we place a target spin-up at the input port, the resulting state $|\uparrow G_+ \downarrow\rangle \equiv |\uparrow\rangle_{\text{in}}|1\rangle_{\text{gate}}|\downarrow\rangle_{\text{out}}$ will have the energy $\tilde{\lambda}_{\uparrow G_+ \downarrow} = -J_2(1 - \frac{1}{2}\Delta)$ which differs significantly from the energies $\tilde{\lambda}_{\downarrow\downarrow\uparrow} = J_2(1 + \frac{1}{2}\Delta) + J_1\Delta$ and $\tilde{\lambda}_{\downarrow\uparrow\uparrow\downarrow} = -J_2(1 + \frac{3}{2}\Delta) + J_1\Delta$ of the states to which it can couple via a single spin-exchange (assuming $\Delta \neq 0$, see below and Supplementary Note 2), see Fig. 1c. Therefore, such an initial state will remain stationary and the control spin excitation on the gate will block the transfer of the target spin between the input and output ports, see Fig. 1d. Indeed, the large energy difference $\tilde{\lambda}_{\uparrow\downarrow\downarrow\uparrow} - \tilde{\lambda}_{\uparrow G_+ \downarrow} \sim |J_2| \gg |J_1|$ will preclude the escape of the gate spin excitation to the output port, while the large energy mismatch $\tilde{\lambda}_{\uparrow G_+ \downarrow} - \tilde{\lambda}_{\downarrow\uparrow\uparrow\downarrow} \simeq 2\Delta J_2$ will suppress the probability of the second-order exchange of spin excitation between the input and

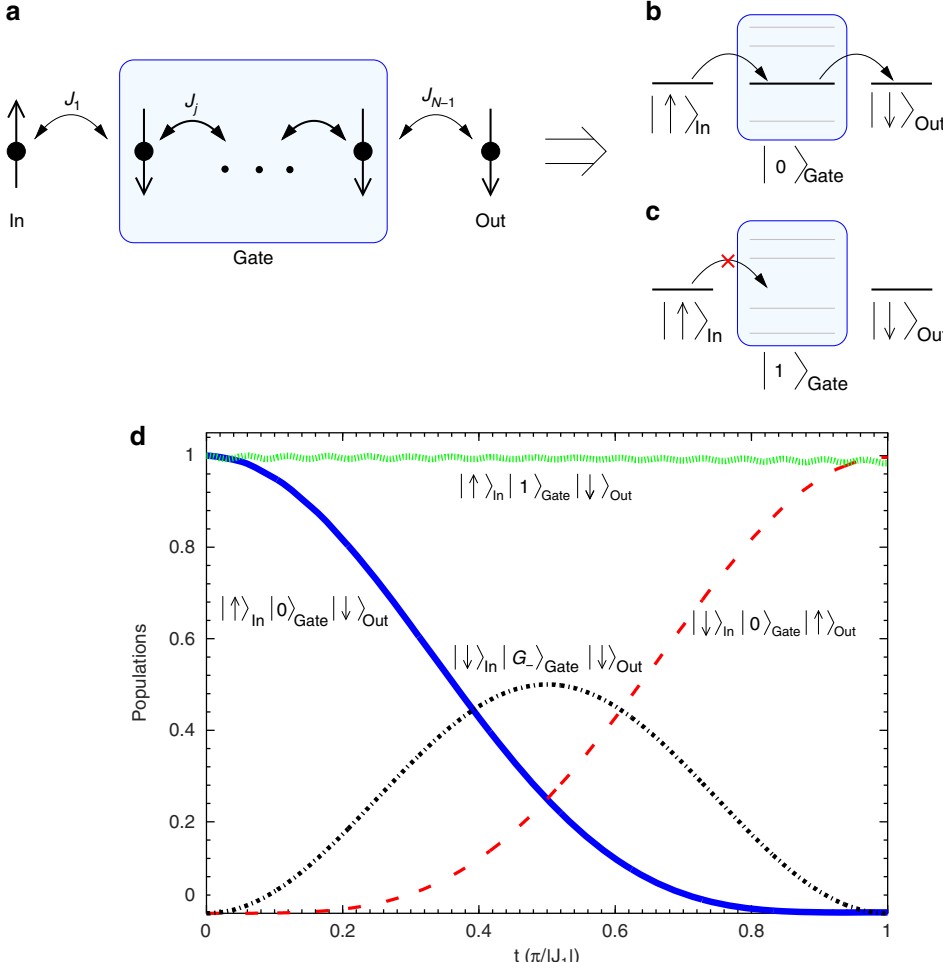

**Figure 1 | Quantum spin transistor implementation in a spin chain.** (**a**) The 'in' and 'out' ports are coupled with $J_{1,N-1}(J_{1,N-1}/J_j \ll 1)$ to the central 'gate' region. (**b**) The energy levels of the gate, split by $\sim J_j/(N-2)$, are additionally tuned by $h_j$, so as to realize resonant transfer of a spin excitation between the input and output ports when the gate is empty, $|0\rangle_{\text{gate}}$. (**c**) The spin excitation transfer is blocked when the gate contains an excited (control) spin, $|1\rangle_{\text{gate}}$. (**d**) Dynamics of an $N=4$ spin system with the open or closed gate. The parameters are $\Delta = -1$, $h = h_- = 0$ and $J_2/J_1 = 17.484$, as obtained for a system of four atoms in the triple-well potential from equation (2) ($V_0 = 500\varepsilon$ and $U = 200\varepsilon$).

output ports by a factor of $\frac{\pi^2}{4}(J_1/J_2)^2 \ll 1$ (with $\Delta = -1$, see below and Supplementary Note 2). Exactly the same arguments apply to the initial state $|\uparrow\, G_-\, \downarrow\rangle$ with the magnetic field set to $h = h_+$.

In the same spirit, we can construct spin transistors with longer spin chains (see Supplementary Note 3 for $N = 5$), the above case of $N = 4$ being the shortest and simplest one. The general idea illustrated in Fig. 1a–c is as follows: The gate region consists of $N - 2$ strongly coupled spins, $|J_j| \gg |J_{1,N-1}|$ for all $j \in \{2, \dots, N-2\}$. Therefore, the single excitation space of the gate has $N - 2$ eigenstates $|G_i\rangle$ split by $\delta\lambda_G \sim \frac{4J_j}{N-2}$. With the magnetic field $h_j$, we tune one of these eigenstates, say $|G_{i'}\rangle$, in resonance with the single excitation input $|\uparrow\downarrow \dots \downarrow\downarrow\rangle$ and output $|\downarrow\downarrow \dots \downarrow\uparrow\rangle$ states. Assuming $|J_{1,N-1}| \ll |\delta\lambda_G|$, all the other eigenstates $|G_i\rangle$ will remain decoupled during the transfer, and we will have a simple three-level dynamics for a single target spin. Note that if the resonant eigenstate $|G_{i'}\rangle$ is antisymmetric, the output state $|\psi\rangle = \alpha|\downarrow\rangle + \beta|\uparrow\rangle$ will be the same as the input state, independent on the sign of $J_j$ (see also Supplementary Note 2), while if we chose symmetric resonant state $|G_{i'}\rangle$, we would obtain the output state $|\psi\rangle = \alpha|\downarrow\rangle - \beta|\uparrow\rangle$ which would be identical to the

input state upon the application of the Pauli-Z quantum gate ($\hat{\sigma}_z$ operation) to the output port. To close the transistor gate, we place a single spin excitation in one of the gate eigenstates $|G_{i \neq i'}\rangle$ from where the control spin cannot leak out since this eigenstate is non-resonant. Simultaneously, the target spin cannot enter the gate region since the double-excitation subspace, to which it is coupled, is shifted in energy due to the spin–spin interaction, resulting in the transfer blockade.

**Physical realization.** A possible system to realize the spin chain Hamiltonian of equation (1) is a cold ensemble of $N$ strongly interacting atoms in a one-dimensional trap of appropriate geometry[33–36]. Recent experiments have confirmed that spin chains may indeed be realized this way[37] with tunable microscopic optical traps[38]. An outline of the procedure to map this system onto the Heisenberg *XXZ* spin model is presented in Supplementary Note 4. A pair of suitable internal atomic states can serve as the spin-up and spin-down states, with the microwave or radiofrequency transition frequency further tunable through a global magnetic field inducing spatially-uniform

Zeeman shifts and by tightly focused non-resonant laser beams inducing site-selective ac Stark shifts [31,39]. In the full model, the strong contact interactions between the atoms are described by the dimensionless coefficients $g_{\uparrow\downarrow} \equiv g \gg 1$ and $g_{\uparrow\uparrow} = g_{\downarrow\downarrow} = \kappa g$, where the parameter $\kappa > 0$ is related to the asymmetry parameter of the effective Heisenberg spin-$\frac{1}{2}$ model as $\Delta = \left(1 - \frac{2}{\kappa}\right)$. In turn, the exchange constants of the Heisenberg model $J_j = -\frac{\alpha_j}{g}$ are proportional to the geometric factors $\alpha_j$ which are determined by the single particle solutions of the Schrödinger equation in a one-dimensional confining potential $V(x)$. Hence, the shape of the trapping potential can be used to tune the necessary parameters of the effective spin chain [34].

To realize the Hamiltonian (1) for $N = 4$ particles with $J_{1,3}/J_2 \ll 1$, we use a triple-well potential. This can be modelled in numerous ways, and we choose the physically simple form

$$V(x) = -V_0[e^{-a(x-x_0)^2} + e^{-a(x+x_0)^2}] - Ue^{-bx^2}, \quad (2)$$

shown in Fig. 2a (see the caption for the parameters $a$, $b$ and $x_0$). The potential consists of a shallow Gaussian well at the center and a pair of deeper and narrower wells next to the boundaries. The four lowest-energy single particle wavefunctions of potential (2) are also shown in Fig. 2a. The two lower energy states are nearly degenerate and the corresponding wavefunctions have sizable amplitudes at the deep wells near the boundaries, while the two higher energy states have much larger energy separation, with the amplitudes of the corresponding wavefunctions being large in the shallow well in the middle. Accordingly, the effective exchange interactions satisfy $J_1 = J_3$ and $J_1/J_2 \ll 1$. The dependence of the ratio $J_2/J_1$ on the parameters $V_0$ and $U$ in equation (2) are shown in Fig. 2b,c.

The nearly perfect transfer, or complete blockade, of the target spin for the open, or closed, gate as shown in Fig. 1d was obtained for the system parameters as in Fig. 2a. It is however important to quantify the sensitivity of the spin transistor to uncontrolled fluctuations of the parameters. In Fig. 3a we show the fidelities $F(t_{\text{out}}) = |\langle\downarrow\downarrow\downarrow\uparrow|e^{-iHt_{\text{out}}}|\uparrow\downarrow\downarrow\downarrow\rangle|^2$ of transfer at time $t_{\text{out}} = \pi/|J_1|$ versus the amplitude of random noise affecting the trapping potential or the (effective) magnetic field. We observe that coherent transfer is quite robust with respect to moderate variations in $V_0$, but is rather sensitive to small variations in $U$ and $h$ since they detrimentally affect the gate resonant conditions. Importantly, the gate blockade is virtually unaffected by

uncertainties in $U$, $V_0$, $h$ since it relies on the (large) energy mismatch. In Fig. 3b, we show the dependence of the blockade fidelity $\bar{F}(t_{\text{out}}) = |\langle\uparrow G_+ \downarrow|e^{-iHt_{\text{out}}}|\uparrow G_+ \downarrow\rangle|^2$ on $\kappa$ which determines the asymmetry parameter $\Delta$ of the $XXZ$ model. Clearly, the spin transistor cannot operate when $\Delta = 0$, that is, in the absence of the spin–spin $\hat{\sigma}_z^j \hat{\sigma}_z^{j+1}$ interactions, since then the control spin can leak out of the gate, as mentioned above and in Supplementary Note 2.

## Discussion

To summarize, we have presented a scheme for a quantum spin transistor realized in a Heisenberg spin chain and proposed and analysed its physical implementation with cold trapped atoms. In our scheme, the presence $|1\rangle_{\text{gate}}$ or absence $|0\rangle_{\text{gate}}$ of a control spin excitation in the gate can block or allow the transfer of an arbitrary target spin state between the input and output ports. If the gate is prepared in a superposition of open and closed states, then the initial state of the system with the target spin-up at the input port will evolve at time $t_{\text{out}}$ into the spatially entangled state.

$$|\uparrow\rangle_{\text{in}} \frac{1}{\sqrt{2}}(|0\rangle_{\text{gate}} + |1\rangle_{\text{gate}})|\downarrow\rangle_{\text{out}}$$
$$\rightarrow \frac{1}{\sqrt{2}}(|\downarrow\rangle_{\text{in}}|0\rangle_{\text{gate}}|\uparrow\rangle_{\text{out}} + |\uparrow\rangle_{\text{in}}|1\rangle_{\text{gate}}|\downarrow\rangle_{\text{out}}). \quad (3)$$

Furthermore, if the gate is integrated into a larger system in which the excited spins from the source can be fed (one-by-one or one after the other) into the input port, and the output port is connected to the initially unexcited drain, then the initial gate superposition state will result in a (macroscopically) entangled Schrödinger cat-like state of many spins,

$$|\uparrow\uparrow \cdots \uparrow\rangle_{\text{in}} \frac{1}{\sqrt{2}}(|0\rangle_{\text{gate}} + |1\rangle_{\text{gate}})|\downarrow\downarrow \cdots \downarrow\rangle_{\text{out}}$$
$$\rightarrow \frac{1}{\sqrt{2}}(|\downarrow\downarrow \cdots \downarrow\rangle_{\text{in}}|0\rangle_{\text{gate}}|\uparrow\uparrow \cdots \uparrow\rangle_{\text{out}}$$
$$+ |\uparrow\uparrow \cdots \uparrow\rangle_{\text{in}}|1\rangle_{\text{gate}}|\downarrow\downarrow \cdots \downarrow\rangle_{\text{out}}). \quad (4)$$

We note that with the atomic realization of spin chains, with the spin-up and spin-down states corresponding to the hyperfine (Zeeman) sublevels of the ground electronic state, the preparation of the closed gate state or the coherent superposition of

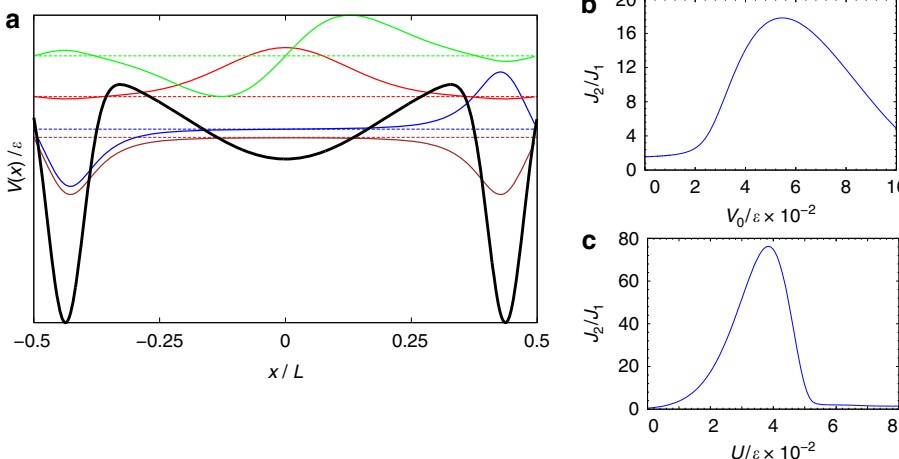

**Figure 2 | Implementation in a triple-well potential. (a)** Shape of the potential in equation (2) (thick solid black curve), and single particle eigenfunctions (thinner solid curves) corresponding to the four lowest-energy levels (dashed horizontal lines), for $V_0 = 500\varepsilon$ and $U = 200\varepsilon$, where $\varepsilon = \frac{\hbar^2}{mL^2}$ with $m$ the atom mass. **(b)** The ratio of exchange coefficients $J_2/J_1$ as a function of $V_0$, for $U = 200\varepsilon$. **(c)** The ratio of exchange coefficients $J_2/J_1$ as a function of $U$, for $V_0 = 500\varepsilon$. Other parameters in equation (2) are $a = \frac{384}{L^2}$, $b = \frac{64}{5L^2}$, $x_0 = \frac{7L}{16}$.

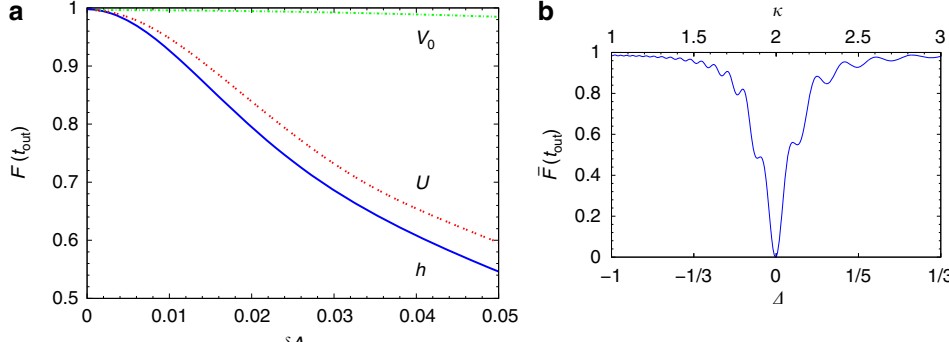

**Figure 3 | Spin transistor sensitivity to fluctuations.** (**a**) Transfer fidelity $F(t_{out})$ averaged over 100 independent realizations of the spin chain. In each realization, one of the parameters in $A = \{V_0, U, h\}$ is a random variable with the Gaussian probability distribution around the ideal mean $A_0$ ($V_0 = 500\,\varepsilon$, $U = 200\,\varepsilon$, $h = h_-$ or $h_+$), while the other two parameters are kept constant. The s.d. is $\sigma_A = A_0 \delta A$ and we take $\sigma_h = h_+ \delta h$ for both $h = h_\pm$ with nearly identical results. (**b**) The blockade fidelity $\bar{F}(t_{out})$ for different values of $\kappa$ (top horizontal axis) or $\Delta$ (bottom horizontal axis).

states $|0\rangle_{gate} \equiv |\downarrow\downarrow\rangle$ and $|1\rangle_{gate} \equiv |G_+\rangle$ (assuming $h = h_- = 0$) can be accomplished by applying to $|\downarrow\downarrow\rangle$ a microwave or two-photon (Raman) optical pulse of proper area ($\pi$ or $\pi/2$) and the frequency matching the energy difference $\delta\lambda = 2J_2\Delta$ between $|\downarrow\downarrow\rangle$ and $|G_+\rangle$. In turn, the initialization of the target spin state at the input port can also be done with resonant microwave or radiofrequency field(s), upon shifting the transition frequency of the atom by a focused laser[31,39], while the readout at the output port can be done by internal state selective fluorescence using the quantum gas microscope set-up[40,41]. Both the initialization and the readout can be accomplished during time intervals $\sim 10\,\mu s$—short compared with $\pi/|J_1| \geq 1 - 10\,ms$ transfer time[39].

**Data availability.** The data that support the findings of this study are available from the corresponding authors upon request.

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

## Acknowledgements

This work was funded in part by the Danish Council for Independent Research DFF Natural Sciences and the DFF Sapere Aude programme, and by the Carlsberg Foundation. A.G.V. acknowledges partial support by Helmholtz Association under contract HA216/EMMI. M.V. is funded by EPSRC EP/M024636/1. D.P. is grateful to the Aarhus Institute of Advanced Studies in Denmark for hospitality and support, and to the Alexander von Humboldt Foundation for support during his stay in Germany.

## Author contributions

O.V.M., A.G.V., M.V., D.P. and N.T.Z. devised the project. O.V.M. and A.G.V. developed the formalism under the supervision of D.P. and N.T.Z. The numerical calculations were carried out by O.V.M. The initial draft of the paper was written by O.V.M., N.T.Z. and D.P. All authors contributed to the revisions that led to the final version.

## Additional information

**Competing financial interests:** The authors declare no competing financial interests.

**How to cite this article**: Marchukov, O. V. *et al.* Quantum spin transistor with a Heisenberg spin chain. *Nat. Commun.* **7**, 13070 doi: 10.1038/ncomms13070 (2016).

