## [Peer review file · Nature Communications]

Reviewers' comments:

Reviewer #1 (Remarks to the Author):

The authors present a scheme for a quantum spin transistor, where the presence or absence of a spin excitation in the gate controls the transfer of a quantum state from the input to the output. The gate can be in quantum superposition, allowing for the generation of spatial quantum entanglement. They also study a concrete realization of their proposal in a cold atom system and its robustness to experimental imperfections. The conclusions are valid and the presentation is good, but I do not find the work sufficiently innovative or impactful to be published in this journal. Quantum communication through spin chains has been studied extensively (see the review article S. Bose, *Contemporary Physics*, 48:1, 13-30 (2007) for references). The novel aspect of this work is the conditional state transfer. This is certainly an interesting twist, but I do not think it elevates the work to the level suitable for publication in *Nature Communications*.

Reviewer #2 (Remarks to the Author):

In 'Quantum spin transistor with a Heisenberg spin chain', Marchukov et al. describe a scheme for creating a transistor that can controllably transmit or block a quantum state being injected in one end of a Heisenberg spin chain. Whereas the paper is purely theoretical, it is written in an accessible fashion that is pleasant to read, leaving most of the technical details to the supplementary material. Below I list a number of comments that may improve the manuscript. However, I would first like to address the central question if the work in its current form is suitable for publication in *Nature Communications*.

In the past decade there have been many proposals for quantum information components. What in my opinion makes this particular work unique is that it is elegant, and universal in the sense that it is based on the very general notion of a spin-1/2 Heisenberg XXZ chain: a system that can in principle be realized in various experimental settings. The idea is so simple that I am almost surprised that it hasn't been proposed before (at least, not to my knowing), and that simplicity of course adds to the impact of the work.

But in this apparent universality lies also a possible weakness. What I mean is that if the selling point of the story is that it can be easily implemented in various real systems, then the authors should be able to truly back that up. The authors describe a specific way to create an XXZ chain using ultra-cold atoms, but in this discussion I miss at least three important issues:

1. How do the authors envisage bringing the input quantum state to the chain, and properly reading out the state on the other side?
2. What about the timing of the gate signal? If I understand correctly, when the gate is left open the quantum state will continuously oscillate from input to output and back. So once opened, the gate should be closed again exactly after a time π/J_1 . The authors should give some estimation of this timescale and a discussion on how this can be implemented.
3. In order for the scheme to work, the chain needs an exactly tuned local magnetic field that is present only on the inner atoms of the chain (the gate atoms), but not on the input and output atoms. At least for the experimental systems that I am familiar with, this is extremely difficult to realize. Do the authors have any solution for this?

In my opinion, consideration of this manuscript for *Nature Communications* should depend on whether the above issues can be properly addressed.

In addition I have some smaller comments and questions:

1. The authors list a number of STM papers describing spin chains built through atom manipulation. In this context I would like to bring the following recent paper to their attention, in which actual spin-1/2 XXZ Heisenberg chains were created: Toskovic et al., Nature Physics (2016), DOI 10.1038/nphys3722.
2. After the header 'Coherent spin transistor', the XXZ chain is suddenly presented without any motivation or introduction. I think it would be good to start here with a sentence along the lines of 'The functionality of the proposed transistor depends on the ability to tune the gate such that one of the intermediate states is coherent with the input and output states, and the other is not. A system that offers the right set of parameters to realize this level of control is the spin-1/2 XXZ Heisenberg chain in a longitudinal magnetic field'.
3. Does the interaction need to be ferromagnetic, or could it also work with antiferromagnetic J (in an odd-numbered chain)?
4. I'm confused about the nomenclature. On page 4, the authors define $|0\rangle_{\text{gate}} = |dd\rangle$ (on the inner two atoms). So $|G\rangle$ represents a single excitation onto $|0\rangle_{\text{gate}}$. But on page 5 they define $|1\rangle_{\text{gate}} = |G+\rangle$. Shouldn't $|1\rangle_{\text{gate}}$ be defined as $|uu\rangle$, and $|G+\rangle$ the symmetric linear combination of $|ud\rangle$ and $|du\rangle$? Otherwise it is not consistent with the definition of $|0\rangle_{\text{gate}}$.
5. I believe there is an important typo on page 4 in the definition of $G+/-$. The second state should be $|du\rangle$ instead of $|ud\rangle$. There are additional typos on page 2: "a novel scheme" and "focus on one such".
6. The blocking of the gate relies on the fact that the symmetric intermediate state is not resonant with the input and output states. But what about energy uncertainty? Can't the transition happen through a virtual process? Perhaps the authors can briefly elaborate on this possibility.
7. The figures need some work. The numbering is strange (b_0, b_1), and in general I think the scheme could be clarified more e.g. by showing the initial, intermediate and final states in Fig. 1b in terms of up/down arrows. Fig. 2 is not very clear to me, but perhaps this makes more sense to people in the community of ultra-cold atoms.
8. Very minor: I don't like the word 'ostensibly' on page 4. There is no discussion that $N=3$ would be the shortest possible chain to accommodate a gate. The discussion is about whether this gate would work, but that's not what 'ostensibly' refers to.

Reviewer #3 (Remarks to the Author):

In this paper, the authors examine the time evolution of several states for some few-site spin-1/2 XXZ Heisenberg chains and demonstrate that the considered spin-chain systems can realize conditional transfer of a spin state at the input (first) site to the output (last) site. A state at the input site is coherently transferred to the output site or is completely blocked depending on the state of the gate (i.e., the inner strongly interacting spins) and the the state of the gate can be tuned by an external magnetic field.

Furthermore, the authors discuss how to realize this spin-chain transistor. They utilize their previous work on engineering spin-chain models using strongly interacting atomic gases (Refs.~32, 33) and suggest how to choose the trapping potential shape to get the necessary parameters of the effective spin-1/2 XXZ Heisenberg chain of $N=4$ sites. They also perform some calculations to check the sensitivity of the spin-chain transistor to uncontrolled fluctuations of the parameters of the suggested triple-well trapping potential. Finally, the authors notice that

their proposal can be also used for preparing entangled states.

These major statements of the paper, in my opinion, are novel and interesting. They may influence further development in this field, in particular, they may stimulate seeking for other realizations of the spin-chain transistor.

The paper is well written and (together with supplementary materials) contains all necessary details to reproduce the results. Although there are some small shortcomings (e.g., Fig. (a) \to Fig. 1(a), page 4, etc), these trifles can be easily removed during the publication process.

In balance, I think I may support publication of this paper in Nature Communications.

=====
Reply to Reviewers' comments:
=====

> Reviewer #1 (Remarks to the Author):

>

> The authors present a scheme for a quantum spin transistor, where
> the presence or absence of a spin excitation in the gate controls
> the transfer of a quantum state from the input to the output.
> The gate can be in quantum superposition, allowing for the generation
> of spatial quantum entanglement. They also study a concrete realization
> of their proposal in a cold atom system and its robustness to
> experimental imperfections. The conclusions are valid and the
> presentation is good, but I do not find the work sufficiently
> innovative or impactful to be published in this journal.
> Quantum communication through spin chains has been studied extensively
> (see the review article S. Bose, Contemporary Physics, 48:1, 13-30
> (2007)
> for references). The novel aspect of this work is the conditional state
> transfer. This is certainly an interesting twist, but I do not think
> it elevates the work to the level suitable for publication in
> Nature Communications.

We appreciate that the Referee found our work novel and interesting and the presentation good. But we respectfully disagree with the Referee that the idea of a quantum spin transistor and its realization with a small spin chain is merely "an interesting twist" in the subject of quantum state transfer through spin chains.

There is no doubt that quantum state transfer through spin chains is an important topic as attested by much attention and many publications, including the excellent introductory overview by S. Bose (we fixed the error in that citation). We, and our close collaborators, have also made several contributions to that research field, and we are well aware of its importance for quantum information, e.g., for achieving scalable integrated quantum computers, connectors, etc. But our main idea in the present paper is the quantum state transistor realizing conditional logic element. This spin transistor can be integrated in a longer chain, to provide conditional transmission or reflection of a propagating excitation. But it can also serve as a stand-alone element connecting qubits. Furthermore, the gate part of our transistor is indeed realized by two or more coupled spins, but in principle it can be realized by just a single quantum system with three or more internal states (one transition resonant with the input-output qubit transition, the other non-resonant to store the gate excitation as a switch). Thus, if we try to construct a gate from qubits, we end up with the minimal system of two coupled qubits (with four states), and not in fact necessarily in the 1D chain configuration.

As a proof-of-principle demonstration of the idea, we use a small setup with cold atoms, which realizes the XXZ spin model under the usual experimental conditions. The main reasons for us in choosing this system are that cold trapped atoms are very pure and experimentally versatile systems available in many labs, and that we ourselves have sufficient experience and many contributions in this field. Other realizations of small spin chains for our transistor are also possible, e.g. chain of coupled quantum dots electrostatically defined by metallic gates on top of a 2D electron gas in a semiconductor, phosphorous dopants in isotopically pure Si, or coupled SQUID qubits.

Hence, we believe that our quantum spin transistor is not merely an interesting addition to the field of quantum communication through spin chains, but is a complementary, and perhaps equally compelling idea on its own.

=====

> Reviewer #2 (Remarks to the Author):

> In 'Quantum spin transistor with a Heisenberg spin chain', Marchukov et al.

> describe a scheme for creating a transistor that can controllably transmit

> or block a quantum state being injected in one end of a Heisenberg spin chain. Whereas the paper is purely theoretical, it is written in an accessible fashion that pleasant to read, leaving most of the technical details to the supplementary material. Below I list a number of comments

> that may improve the manuscript. However, I would first like to address the central question if the work in its current form is suitable for publication in Nature Communications.

> In the past decade there have been many proposals for quantum information

> components. What in my opinion makes this particular work unique is that

> it is elegant, and universal in the sense that it is based on the very general notion of a spin-1/2 Heisenberg XXZ chain: a system that can in principle be realized in various experimental settings. The idea is so simple that I am almost surprised that it hasn't been proposed before (at least, not to my knowing), and that simplicity of course adds to the impact of the work.

We thank the Referee, first of all, for appreciating simple yet elegant and potentially useful physics, and for carefully reading our manuscript and providing useful comments and criticism, which we address below.

> But in this apparent universality lies also a possible weakness. > What I mean is that if the selling point of the story is that it > can be easily implemented in various real systems, then the authors > should be able to truly back that up. The authors describe a specific > way to create an XXZ chain using ultra-cold atoms, but in this

> discussion I miss at least three important issues:

It is a correct observation that our coherent spin transistor scheme can in principle be implemented in a variety of systems that can realize small XXZ spin chains (the minimal gate consists of just two strongly coupled spins). Examples include chains of coupled quantum dots electrostatically defined by metallic gates on top of a 2D electron gas in a semiconductor, phosphorous dopants in isotopically pure Si, or coupled superconducting qubits.

As a proof of principle demonstration of the idea, we employ here a small setup of cold atoms, which realizes the XXZ spin model under usual experimental conditions. The main reasons for us in choosing this system are that cold trapped atoms are very pure, tunable and experimentally versatile systems available in many labs, and that we ourselves have sufficient expertise and contributions in this field.

> 1. How do the authors envisage bringing the input quantum state to
> the chain, and properly reading out the state on the other side?

Ideally, the spin transistor will be an integral element of a larger quantum (spintronic or, here, atomtronic) system, and it will serve as a coherent logical element that can be engaged (gate switched on and off) according to a particular computation or simulation algorithm.

In a bare-bone realization and demonstration of this element with cold-atom setup, it will involve just four atoms in an optical trap with the shape as in fig. 2, realized by far off resonant lasers as in, e.g. S. Murmann et al. Phys. Rev. Lett. 114, 080402 (2015). The atoms are prepared initially (optically pumped, for instance) in one of the internal states which corresponds to the spin-down state. The gate state can be switched between the on $|0\rangle$ or off $|1\rangle$ configurations

by a microwave or two-photon Raman pulse matching the transition frequency $\Delta = 2 J_2 \lambda$ (see the last paragraph of the summary and outlook section). This pulse will not affect the input- and output-port spin states because it will not be resonant with the transition resonance of those spins (assumed degenerate in the interaction picture), provided the amplitude (Rabi frequency) of the pulse is small compared to $\delta\lambda$.

In turn, the input spin can be flipped in a way similar to that in T Fukuhara et al, Nature Phys 9, 235 (2013) or T Xia et al., Phys. Rev. Lett. 114 (10), 100503 (2015) [refs. 31-39]. Namely, non-resonant laser tightly focused onto the input spin atom induces relative ac Stark shift of the internal atomic states representing the spin-up and spin-down states. Then a microwave or rf pulse now resonant with only this shifted transition flips the spin. The cross-talk with the neighboring gate spins will be small if the Stark laser is focused to a spot-size not exceeding the interatomic distance, and the mw Rabi is smaller than the Stark shift [as in the above papers].

The read-out of the spin state can be accomplished by spin-selective (internal state selective) fluorescence, as in the quantum gas microscope experiments of M. Greiner group, Bakr et al. Nature 462, 74 (2009), and I. Bloch group, Sherson et al., Nature 467, 68-72 (2010), T Fukuhara et al, Nature Phys 9, 235 (2013) and by now many others (including for fermionic atoms).

We have added a short discussion in the paper to outline these steps and cite relevant papers.

> 2. What about the timing of the gate signal? If I understand correctly,
> when the gate is left open the quantum state will continuously oscillate
> from input to output and back. So once opened, the gate should be closed
> again exactly after a time π/J_1 . The authors should give some estimation
> of this timescale and a discussion on how this can be implemented.

This is a pertinent question which we did not clarify sufficiently in the paper. We indeed deal with unitary dynamics, and in the four-spin system (with open boundary conditions), the open gate correspond to a three-state system in which a single excitation would perform periodic oscillations between the first and the last states via the intermediate resonant state, and the half-period of this oscillations is π/J_1 , as the the Referee correctly pointed out. Hence, while we initialize the input state, the exchange coupling between spins 1 and 2 should be closed, $J_1=0$, then it should be quickly opened and once the state is transferred to spin 4, the exchange coupling between spins 3 and 4 should be closed, $J_3=0$. The strong J_2 can be left unchanged. The time scale for opening and closing the weak coupling should be small compared to $1/J_{\{1,3\}}$. Alternatively, we should be able to initialize and read out much faster than $1/J_{\{1,3\}}$. Typical values for $J_{\{1,3\}}$ are in the 0.1-1KHz range [ref. 34], hence the switching time, or initialization and readout times, should be 100us or less.

One can envision three possible ways to start and stop the exchange process:

- (1) raise the potential barrier between atoms 1-2 or 3-4 by focused lasers to reduce the wavefunction overlap and thereby the exchange interaction;
 - (2) use a uniform magnetic field to suddenly increase g for all the atoms near the Feshbach resonance, and thereby reduce the exchange interaction α/g ; or
 - (3) perhaps the easiest way would just be to apply ac Stark focused laser to the desired atom to detune its spin-transition resonance (as described above for preparation) and thus suppress the exchange interaction and do the initialization and readout.
- All these methods can be applied on a short timescale of 1-10 μ s.

We note parenthetically, that the $J_{\{1,3\}}$ exchange opening-closing does not have to be sudden, in which case t_{out} will not be sharply defined as π/J_1 , but via some effective (three-level) pulse area being π , but this is less intuitive.

We have added a short discussion pertaining to this issue.

Note finally that if the transistor is integrated in a larger chain (see the last section of the paper) than the incoming spin excitation (magnon) is either reflected back to the source, or transmitted to the drain, and no switching on and off of $J_{\{1,3\}}$ is now necessary.

> 3. In order for the scheme to work, the chain needs an exactly tuned
> local magnetic field that is present only on the inner atoms of the
chain
> (the gate atoms), but not on the input and output atoms. At least for
> the experimental systems that I am familiar with, this is extremely
> difficult to realize. Do the authors have any solution for this?

The value of the (effective) magnetic field h applied to the gate spins depends on the parameters J_2 and Δ of the spin chain, and the choice of the resonant state. h can be zero, which is in fact the case we used ($h = h_- = 0$) to demonstrate the operation of the spin transistor in the paper. Our chain consists of 4 spins with parameter corresponding to the usual setup with cold, strongly interacting atoms with large, positive and spin-independent g leading to $\Delta = -1$.

The Referee is right that in general applying local magnetic field affecting just one or a few atoms in a longer 1D chain would be difficult because the typical interatomic separation is of the order of a micrometer. However, a relative ac Stark shift of a pair of atomic internal states can be induced by a non-resonant laser focused on one or a few atoms (see above). This is equivalent to a relative Zeeman shift of the pair of spin states in an effective magnetic field.

In other realizations of spin chains, e.g. chain of coupled quantum dots electrostatically defined by metallic gates in semiconductors, phosphorous dopants in isotopically pure Si, or SQUID qubits, the local control of the energies of the effective spin-up and spin-down states is done via appropriate voltages and currents applied to the conducting elements.

We have now added brief discussion at the beginning of section "Physical realization" to mention the tunability of the trapping potential and local resonance frequencies of the atoms, and cite relevant papers. But again, our prime choice of the resonant gate state assumes zero magnetic field for all atom, in a standard setup for cold, strongly interacting atoms.

> In my opinion, consideration of this manuscript for Nature
Communications
> should depend on whether the above issues can be properly addressed.

We hope we have provided convincing answers to clarify these issues.

> In addition I have some smaller comments and questions:

> 1. The authors list a number of STM papers describing spin chains
> built through atom manipulation. In this context I would like to
> bring the following recent paper to their attention, in which actual
> spin-1/2 XXZ Heisenberg chains were created: Toskovic et al.,
> Nature Physics (2016), DOI 10.1038/nphys3722.

This is of course an interesting and relevant paper which we now cite.

> 2. After the header 'Coherent spin transistor', the XXZ chain is
> suddenly presented without any motivation or introduction. I think
> it would be good to start here with a sentence along the lines of
> 'The functionality of the proposed transistor depends on the ability
> to tune the gate such that one of the intermediate states is coherent
> with the input and output states, and the other is not. A system that
> offers the right set of parameters to realize this level of control
> is the spin-1/2 XXZ Heisenberg chain in a longitudinal magnetic field'.

That is indeed a good suggestion and we have added a text, similar to one proposed by the referee, which provides motivation and smooth transition to the spin chain introduction.

> 3. Does the interaction need to be ferromagnetic, or could it also
> work with anti-ferromagnetic J (in an odd-numbered chain)?

The sign of the exchange couplings J can be positive, or negative ($J \Delta > 0$ is ferromagnetic and < 0 anti-ferromagnetic). The state transfer would work equally well in either case, but the relative phase of the final state should be taken into account.

In the standard state transfer protocols [see e.g. refs 11, 14, 15], upon resonant transfer of the input state $|\psi\rangle = \alpha |d\rangle + \beta |u\rangle$ from site 1 to site N , the output state is given by $\alpha |d\rangle + e^{i\phi} \beta |u\rangle$, where the phase is $\phi = -\text{sign}(J) \pi/2 (N-1) \pmod{2\pi}$ with J the intersite couplings assumed all to have the same sign. Hence, depending on N , the output state for either sign of J is the same as the input to within proper phase rotation of the amplitude of $|u\rangle$, which is a trivial single qubit operation (unity, $\pi/2$ rotation or Pauli σ_z operation).

Here, independent of the chain spin number N , we construct a 3 state system, with the excitation on the input spin, resonant gate state, and output spin.

Now the input state $|\psi\rangle = \alpha |d\rangle + \beta |u\rangle$ is transferred to the output

state via the resonant $|G\rangle$ state which is antisymmetric. This state is coupled to the input spin with amplitude $-J_1/\sqrt{2}$, and to the output spin with amplitude $+J_3/\sqrt{2}$ (see eq. (11) of the supplement).

Recall that $J_1 = J_3$. This means that the transferred state is

$\alpha |d\rangle + \beta |u\rangle$, since the factor multiplying the amplitude of $|u\rangle$ is $(i)(-i)=1$ if $J_{\{1,3\}} < 0$ and the same $(-i)(i)=1$ if $J_{\{1,3\}} > 0$.

Note however, that if we used for the open gate the transition to the symmetric state $|G_+\rangle$ (and $|G_-\rangle$ for the storage of gate spin-up), then both couplings would have the same sign, $-J_1/\sqrt{2}$ and $-J_3/\sqrt{2}$, see eq. (11) SM. Then the output state would have the phase $\phi = -\text{sign}(J) \pi/2 (3-1) = -\text{sign}(J) \pi$. The sign of Δ does not matter.

Hence, the output state for either sign of J would be $\alpha |d\rangle - \beta |u\rangle$, which is the same as the input after applying the Pauli-Z quantum gate to the output port.

We have now added this detailed discussion in the supplementary material, and explicitly mention in the paper the relation between the output and input states and that the sign of J 's does not affect the transfer.

> 4. I'm confused about the nomenclature. On page 4, the authors
> define $|0\rangle_{\text{gate}} = |dd\rangle$ (on the inner two atoms). So $|G_-\rangle$ represents
> a single excitation onto $|0\rangle_{\text{gate}}$. But on page 5 they define
> $|1\rangle_{\text{gate}} = |G_+\rangle$. Shouldn't $|1\rangle_{\text{gate}}$ be defined as $|uu\rangle$, and $|G_+\rangle$ the
> symmetric linear combination of $|ud\rangle$ and $|du\rangle$? Otherwise it is
> not consistent with the definition of $|0\rangle_{\text{gate}}$.

The gate state $|0\rangle_{\text{gate}}$ is indeed $|dd\rangle$. The transition between this state $|0\rangle_{\text{gate}} = |dd\rangle$ and the single (antisymmetric) excitation state $|G_-\rangle = |ud\rangle - |du\rangle$ of the gate is tuned to be resonant with the transition $|u\rangle \rightarrow |d\rangle$ of the "in" spin and also with the transition $|d\rangle \rightarrow |u\rangle$ of the "out" spin. That's why we get the resonant two-step transfer $|u\rangle_{\text{in}} |0\rangle_{\text{gate}} |d\rangle_{\text{out}} \rightarrow |d\rangle_{\text{in}} |G_-\rangle_{\text{gate}} |d\rangle_{\text{out}} \rightarrow |d\rangle_{\text{in}} |0\rangle_{\text{gate}} |u\rangle_{\text{out}}$ with always one excitation in the system.

Next, the gate state $|1\rangle_{\text{gate}} = |G_+\rangle = |ud\rangle + |du\rangle$ corresponds to the symmetric single spin excitation in the gate. This state has energy difference $2J_2$ with state $|G_-\rangle$. Therefore the transition $|G_+\rangle \rightarrow |dd\rangle$ from this state is non-resonant with the transition $|d\rangle \rightarrow |s\rangle$ of the in and out spins. Therefore the symmetric single spin excitation of the gate $|1\rangle_{\text{gate}} = |G_+\rangle$ cannot escape. Simultaneously, the transition $|G_+\rangle \rightarrow |uu\rangle$ to the gate double excitation state is nonresonant with the transition $|u\rangle \rightarrow |d\rangle$ of the in spin, due to the Z-Z spin interaction. That's why the second spin excitation from the input port cannot enter the gate. So the transfer is blocked.

To summarize, for the magnetic field $h=h_- = 0$, our open gate corresponds to the initial state state $|0\rangle = |dd\rangle$, and we use the resonant state $|G_-\rangle$ to transfer the spin excitation via it. On the other hand, our closed gate corresponds to non-resonant state $|G_+\rangle$, which we denote as $|1\rangle$ (we now explicitly say it in the paper).

> 5. I believe there is an important typo on page 4 in the definition
> of G_{\pm} . The second state should be $|du\rangle$ instead of $|ud\rangle$. There are
> additional typos on page 2: "a novel scheme" and "focus on one such".

That's correct, G_{\pm} are the symmetric and antisymmetric combinations
of $|du\rangle$ and $|ud\rangle$, and we fixed that typo.

An on page 2 we also corrected the "a" and added the missing "on".

> 6. The blocking of the gate relies on the fact that the symmetric
> intermediate state is not resonant with the input and output states.
> But what about energy uncertainty? Can't the transition happen
> through a virtual process? Perhaps the authors can briefly elaborate
> on this possibility.

This is an important question which of course should have been
clarified in the paper and supplementary material.

The states $|u\rangle_{\text{in}} |1\rangle_{\text{gate}} |d\rangle_{\text{out}}$ and $|d\rangle_{\text{in}} |1\rangle_{\text{gate}} |u\rangle_{\text{out}}$ have
the same energy, but they are not directly connected. Instead,
they are connected via spin exchange to the intermediate
states $|d\rangle_{\text{in}} |uu\rangle_{\text{gate}} |d\rangle_{\text{out}}$ and $|u\rangle_{\text{in}} |dd\rangle_{\text{gate}} |u\rangle_{\text{out}}$
which are non-resonant, detuned by $\sim 2 \Delta J_2$ and $2 J_2$,
assuming $J_1 \ll J_2$, see the discussion on p.5 and supplementary.
So in principle one can envisage resonant second-order transition
 $|u\rangle_{\text{in}} |1\rangle_{\text{gate}} |d\rangle_{\text{out}} \rightarrow |d\rangle_{\text{in}} |1\rangle_{\text{gate}} |u\rangle_{\text{out}}$ via these
non-resonant intermediate states. The amplitude of these
second-order processes is $\sim (J_1/\sqrt{2})^2/(2J_2) \ll J_1$ ($\Delta = -1$),
which, due to our assumption $J_1 \ll J_2$, is very small.
During the open-gate transfer time π/J_1 , the probability
for this blocked transition to occur is then
 $|\pi/J_1 J_1^2/(2J_2)|^2 = (\pi/2)^2 (J_1/J_2)^2 \ll 1$.

We now clarify this issue in the paper, and present more details
in the supplemental.

Since the gate blockade relies on the large energy mismatch, it
is not sensitive to small fluctuations of the system parameters,
see. Fig. 3 inset. We now explicitly mention this in the text.

> 7. The figures need some work. The numbering is strange (b0, b1),
> and in general I think the scheme could be clarified more e.g. by
> showing the initial, intermediate and final states in Fig. 1b in
> terms of up/down arrows. Fig. 2 is not very clear to me, but perhaps
> this makes more sense to people in the community of ultra-cold atoms.

We thought that the numbering (b0), (b1) would be intuitive since
it corresponds to the gate state 0 and 1. This is of course not
critical and we would leave it to the Editor's discretion to keep
it that way or change it -- if it hopefully comes to that.

We have added in Fig. 1(c) the curve for the dynamics of the
intermediate state, along with the initial and the final, and

modified the state labels to be consistent with the ones in the text. This now illustrates the time evolution of the three relevant states of the system.

Fig. 2 shows the overall potential and four lowest single particle wavefunctions from which we construct the spin chain and their energies. The two lowest-energy wavefunctions have nearly degenerate energies and are localized at the deep wells, which translates into small exchange couplings $J_{\{1,3\}}$ (cf last section of the supplemental), while the higher energy wavefunctions are mostly in the shallow middle well, and the large energy splitting corresponds to strong exchange coupling J_2 . We have slightly expanded the last paragraph of the supplemental to mention these issues which are derived and discussed in detail in refs. 32-36.

- > 8. Very minor: I don't like the word 'ostensibly' on page 4.
- > There is no discussion that $N=3$ would be the shortest possible
- > chain to accommodate a gate. The discussion is about whether
- > this gate would work, but that's not what 'ostensibly' refers to.

We agree, perhaps 'ostensibly' is not the most appropriate word here, and we have changed it to the simple "One might think that" We discuss the $N=3$ case in the supplemental, and show that even though this is the shortest possible spin chain to accommodate the "in", "gate" and "out" elements, it does not work because the gate excitation would escape.

=====

> Reviewer #3 (Remarks to the Author):

> In this paper, the authors examine the time evolution of several
> states for some few-site spin-1/2 $\text{\$XXZ\$}$ Heisenberg chains and
> demonstrate that the considered spin-chain systems can realize
> conditional transfer of a spin state at the input (first) site
> to the output (last) site. A state at the input site is coherently
> transferred to the output site or is completely blocked depending on
> the state of the gate (i.e., the inner strongly interacting spins)
> and the the state of the gate can be tuned by an external magnetic
field.

> Furthermore, the authors discuss how to realize this spin-chain
transistor.
> They utilize their previous work on engineering spin-chain models using
> strongly interacting atomic gases (Refs.~32, 33) and suggest how to
choose
> the trapping potential shape to get the necessary parameters of the
> effective spin-1/2 $\text{\$XXZ\$}$ Heisenberg chain of $\text{\$N=4\$}$ sites. They also
> perform some calculations to check the sensitivity of the spin-chain
> transistor to uncontrolled fluctuations of the parameters of the
suggested
> triple-well trapping potential. Finally, the authors notice that their

- > proposal can be also used for preparing entangled states.
- > These major statements of the paper, in my opinion, are novel and
- > interesting. They may influence further development in this field,
- > in particular, they may stimulate seeking for other realizations
- > of the spin-chain transistor.
- > The paper is well written and (together with supplementary materials)
- > contains all necessary details to reproduce the results. Although there
- > are some small shortcomings (e.g., Fig. (a) \rightarrow Fig. 1(a), page 4,
- > etc),
- > these trifles can be easily removed during the publication process.
- > In balance, I think I may support publication of this paper
- > in Nature Communications.

We thank the Referee for the concise summary showing understanding and appreciation of our work, and for recommending its publication.

We have corrected several errors and misprints (but the occasional missing of figure numbers seems to be a bug of the latex nature style which we tried to fix).

=====
 Summary of changes
 =====

We corrected several minor errors and misprints in the text and formulas that we found, and also the ones indicated by the referees, and made minor changes to improve the text.

In the abstract we slightly modified and expanded the last sentence, to better emphasize that the cold-atom realization of our spin transistor scheme would be proof-of-concept, but our scheme is generally applicable to many other implementations of tunable spin chains.

In the beginning of sec. "Coherent spin transistor" we added new text suggested by Referee 2 to motivate the following spin chain Hamiltonian and its properties.

On page 4, we have replace the "Ostensibly" with a more appropriate "One might think that"

On page 5, we have corrected the definition of the gate eigenstates $|G_{\pm}\rangle$

The paragraph on pp. 5-6 on the closed gate (starting with "Next we place a single excitation...") was expanded to clarify the issue of suppression of non-resonant transfer (more details are now given in the supplementary, see below)

In the last paragraph of section "Coherent spin transistor" we

added a new sentence "Note that if the resonant eigenstate ... " to clarify the issue of the sign of the spin-up amplitude of the transferred state, which depends on the choice of the resonant gate state, but not on the sign of the exchange couplings J (more details are now given in the supplementary, see below)

The first paragraph of section "Physical realization" was modified and expanded to mention the tunability of the optical trapping potential and local resonance frequencies of the atoms, and cite relevant papers.

The last paragraph of section "Discussion" was modified and expanded to further discuss the experimental issues pertaining to preparation and detection of the system of cold trapped atoms.

We have modified Fig. 1(c) to include the curve for the evolution of the intermediate resonant state, and adapted labels to match the definitions in (b0,b1) and the main text.

In Fig. 3 the caption was slightly modified, to make it clear how we take the standard deviation for $h=h_-$ (which is zero) and that for both possible values of h we get nearly identical results.

Ref. 12. Bose, S. Quantum Communication through Spin Chain Dynamics: an Introductory Overview, Contemporary Physics 48, 13-30 (2007) was corrected (the page and year).

We have added new references:

27 on atomic spin-chain realization, mentioned by the Referee
39 (8 in the supplement) on microscopic versatile optical traps
40 on selective manipulation of single atoms with lasers and MWs in dense arrays of optical traps
41, 42 on quantum gas microscope for single-atom fluorescence detection

We have slightly expanded and improved the supplemental material.

In section "N=3 spin chain", after eq. (7) we added a discussion pertaining to the sign of the amplitude of the transferred state.

This is then used in section "N=4 spin chain" in the newly added texts on pp 10 and 11 to clarify whether or not the amplitude of the transferred state will depend on the choice of the resonant gate state, but not the sign of the exchange couplings.

Close to the end of section "N=4 spin chain", after eqs. (27) we added a new text to discuss the second order transfer process and its suppressed probability.

At the end of section "Implementation of the XXZ spin chain ...", after eq. (46) we added a sentence "Such a potential can be realized in an optical trap using appropriately focused, far-off-resonant

laser beams [8]".

In the next paragraph, we added a new text to outline the physical meaning and consequences of the single-particle eigenfunctions shown in Fig. 2 of the main text.

Finally, we have added section naming (Introduction, Results, Discussion) according to the standards of Nature Communications.

REVIEWERS' COMMENTS:

Reviewer #2 (Remarks to the Author):

The authors did a very extensive job in addressing the issues that I raised in the first review round. Most notably, they managed to take away my biggest concern, which was the requirement for having a local magnetic field: as the authors explained clearly in their response, by a proper choice of system parameters this local field can also be zero. I had not realized this in the first round.

From the other two reviewers there was only one concern, voiced by reviewer 1, that the conditionality of this gate would not be a sufficient novelty. This criticism was not well supported with arguments and I fully agree with the authors' rebuttal that conditionality is in fact a crucial step and worthy of publication at this level.

For the above reasons I support publication in Nature Communications at this stage. However, I add to this statement that I have an experimental background and have reviewed the paper primarily from a general physics interest perspective. I feel that I am not in a position to properly judge the technical contents of the calculations, and I trust that the other two reviewers (who are presumably more knowledgeable in this particular field) have carefully verified the scientific soundness of this work.